# Exercise and/or Genistein Do Not Revert 24-Week High-Fat, High-Sugar Diet-Induced Gut Microbiota Diversity Changes in Male C57BL/6J Adult Mice

**DOI:** 10.3390/microorganisms10112221

**Published:** 2022-11-10

**Authors:** Carmen P. Ortega-Santos, Layla Al-Nakkash, Corrie M. Whisner

**Affiliations:** 1College of Health Solutions, Arizona State University, Phoenix, AZ 85004, USA; 2Department of Physiology, College of Graduate Studies, Midwestern University, Glendale, AZ 85308, USA

**Keywords:** gut microbiota, genistein, exercise, obesity, western diet, high-fat diet, weight gain

## Abstract

The gut microbiota (GM) has been hypothesized to be a potential mediator in the health benefits of exercise and diet. The current literature is focused on the prevention effects of exercise and diet and could benefit from exploring whether these treatments alone or combined can treat obesity via the gut microbiome. This study aimed to explore the effects of genistein, exercise, and their synergistic effect to revert diet-induced obesity and gut microbiota changes. A total of 57 male adult C57BL/6 mice were randomized to 24 weeks of unpurified diet (chow) or a high-fat, high-sugar diet (HFD; 60% fat total energy). After the first 12 weeks, animals on the HFD were randomized into: HFD + chow, HFD, HFD + exercise (HFD + Exe), HFD + genistein (HFD + Gen), and HFD + Exe + Gen. We compared the body weight change between groups after 24 weeks. GM (α-diversity and ß-diversity) was profiled after sequencing the 16S rRNA gene by Illumina MiSeq. HFD + Exe + Gen significantly (*p* < 0.05) decreased weight gain relative to the HFD with only HFD + chow reverting the body weight change to that of chow. All diets including HFD reduced the GM richness (observed amplicon sequence variants) relative to chow with the HFD + Gen and HFD + Exe resulting in significantly lower phylogenetic diversity compared to the HFD. Data did not support an additive benefit to the GM for HFD + Gen + Exe. HFD + Exe + Gen showed a greater capacity to revert diet-induced obesity in adult male mice, but it was not as effective as switching from HFD to chow. Lifestyle treatment of HFD-induced obesity including exercise and genistein resulted in a reduction in weight gain and GM richness, but switching from HFD to chow had the greatest potential to revert these characteristics toward that of lean controls.

## 1. Introduction

The gut microbiota has been hypothesized to be a potential mediator in the health benefits of exercise and diet [1,2]. Although the effects of bioactive dietary compounds (e.g., polyphenols) on the gut microbiota have attracted major attention in the last decade, much less is known about the effects of exercise and the potential synergistic effect of dietary bioactive compounds and exercise on the gut microbiota.

The gut microbiota can metabolize phytoestrogens such as genistein to their bioactive metabolites (e.g., 5-hydroxy-equol). Genistein, the most abundant flavonoid in soy, has been studied for its anti-cancerous effects [3]. However, recent evidence has shown genistein’s anti-obesogenic effects mediated by gut microbiota changes during diet-induced obesity [4,5,6,7,8,9,10] and menopause-associated weight gain using rodent models. Male and female Ob/Ob mice after 4 weeks of genistein supplementation (600 mg/kg of body weight) experienced reductions in weight gain when compared to the standard rodent diet fed mice [8]. Lu et al. tested genistein or a combination of soybean polysaccharides to prevent HFD-induced weight gain and dyslipidemia by modulation of gut microbiota. The combination of soybean polysaccharides was significantly more robust to changing gut microbiota composition, enhancing *Bacteroidetes* relative abundance, preventing dyslipidemia, and preventing body weight gain relative to HFD-only fed mice after 12 weeks [4]. Ovariectomized rodent models similarly showed that genistein ameliorates weight gain when exposing animals to a high-fat diet [11,12]. Yang et al. [13] tested the capacity of genistein to revert impaired glucose metabolism in type 2 diabetic mice and observed a significant increase in the abundance of *Bacteroidetes* and *Prevotella* and concentrations of short-chain fatty acids. The increase in short-chain fatty acids and reduction in the abundance of *Proteobacteria* are considered hallmarks of a healthy gut microbiota profile and have been associated with anti-obesogenic effects elsewhere [4]. In college female athletes, the supplementation with isoflavone-rich (161 mg of isoflavones, 38 mg of genistein, per serving) protein was not enough to prevent body weight gain (~0.5 kg) after 16 weeks compared to a casein–protein supplement [14], whereas obese male individuals consuming 50 mg genistein/day for eight weeks exhibited a genistein-mediated improvement in insulin sensitivity by increasing gut microbial diversity and increased abundance of *Akkermansia muciniphila* or *Ruminoccocus bromii* [15]. They did not report any genistein effects on body weight or adiposity. Overall, the literature mostly focuses on genistein as obesity preventive rather than as obesity treatment.

Animals have been the primary model for understanding how exercise prevents diet-induced obesity and associated metabolic abnormalities (e.g., diabetes) via the gut microbiota [16,17,18]. *Ruminococcaceae* and *Lachnospiraceae* family members were significantly increased across the studies utilizing higher intensity exercise interventions. Data from Evans et al. [16] suggest a potential association between changes in gut microbiota and body composition. They found that changes in gut microbiota were proportional to the distance run by the mice and the exercised mice experienced reduced weight gain and body fat after 12 weeks of intervention (high-fat diet + forced endurance training). Furthermore, an inverse correlation was found significant between the exercise distance and the change in *Firmicutes* and *Bacteroidetes* relative abundance. Denou et al. showed that exercise trained mice experienced improvements in glucose tolerance and fitness in the absence of body weight differences (compared to controls) during diet-induced obesity intervention [18]. One study in sedentary humans with prediabetes and type 2 diabetes [19] found that exercise had the capacity to change the gut microbiota and reduced whole-body fat percentage and abdominal visceral fat at the end of the intervention. An improvement in endotoxemia was also recorded alongside increased *Bacteroides* and decreased *Blautia* relative abundance after two weeks of exercise. However, no statistical analysis was used to calculate the potential link between both of them. Despite existing literature, more research is needed to know the implications of those changes with health.

Exercise and diet alone suggest promising preventive effects on obesity and associated metabolic abnormalities by changing the gut microbiota towards a more diverse community that also comprises a greater abundance of health-associated bacterial taxa (e.g., *Ruminococcaceae*). Using both lifestyle treatments together may potentiate each other’s effects on the gut microbiota to induce anti-obesogenic effects. Nagano et al. [20] studied the synergistic effect of cellulose fiber (0.2% cellulose fiber in their diet via drinking water) and exercise (forced aerobic exercise) to prevent obesity in a diet-induced murine model. They found the combination of cellulose and exercise to be superior for both body weight management and gut microbiota composition changes. The combined group had significantly greater abundance of *Eubacteriaceae*, whereas the exercise-only group experienced significant increases in *Ruminococcaceae*, as suggested by others [16,17,18]. We have previously shown that feeding a diet supplemented with genistein and aerobic exercise training in adult male and female mice, affords protection against the deleterious effects induced by a high-fat, high-sugar diet on the gut microbiota [21]. Isoflavones and aerobic training showed superior capacity to prevent weight gain in ovariectomized female rats compared to either of the treatments alone [6,22]. While this prior work suggested that genistein and exercise have preventative effects, we now aimed to test if the combination of genistein and exercise could be used as a treatment over a longer period of time. Therefore, the current study aimed to test the capacity of aerobic exercise, genistein, and their combined effect on reverting the deleterious effects (accelerated weight gain and reduced GM diversity) of a high-fat, high-sugar diet on the gut microbiota over 24 weeks in male mice. We hypothesized that the combined treatment would significantly revert the decrease in gut microbial diversity induced by the high-fat, high-sugar diet.

## 2. Materials and Methods

### 2.1. Mice and Exercise/Diet Protocols

Male (*n* = 57) C57BL/6 mice, aged 4–5 weeks, were purchased from Charles River (Wilmington, MA, USA). All procedures were approved by the Institutional Animal Care and Use Committee at Midwestern University in Glendale, AZ, USA. All mice were randomly assigned for 24-weeks to unpurified diet (chow, *n* = 9) or HFD (*n* = 48). After following these diets for 12 weeks, animals in the HFD group were randomized a second time into the following new groups: control (fed unpurified diet, *n* = 10), HFD (*n* = 10), HFD + exercise (HFD + Exe; *n* = 10), HFD + genistein (HFD + Gen; *n* = 8), HFD + Exe + Gen (*n* = 10) (Figure 1). The HFD + Exe and HFD + Exe + Gen groups performed moderate intensity exercise 5 days per week on an electrically driven treadmill (Columbus Instr., Columbus, OH, USA) for a period of 12 weeks. The exercise protocol consisted of a 3-week graded increase in exercise duration and intensity as follows: week 1, 10 min at 10 m/min; week 2, 20 min at 10 m/min; week 3, 30 min at 12 m/min; weeks 4–12, 30 min at 15 m/min. All mice were provided with food (Table 1) and drink *ad libitum* and maintained in a room with an alternating 12 h light/dark cycle maintained at 22 °C with a relative humidity between 30 to 70% as required by the Care and Use of Laboratory Animals to keep terrestrial animals’. Mice were housed together with up to 3 mice per cage differentiated by tail color (red, blue, and black). The HFD (20% carbohydrate, 20% protein, and 60% fat) was purchased from Dyets (Dyets Inc., Bethlehem, PA, USA) for which a complete breakdown of ingredients can be seen in Table 1. The HFD + Gen and HFD + Exe + Gen groups feed was supplemented with genistein at a concentration of 600 mg of genistein/kg of diet. The drinking water in all the groups except control and chow were supplemented with 42 g/L of sweetener, consisting of 55% fructose and 45% sucrose. Body weights and general health were monitored weekly for the entire 24-week study. Animal care was conducted in accordance with established guidelines, and all protocols were approved by the Midwestern University Institutional Animal Care and Use Committee (IACUC project #2880 approved 28 August 2019). 

### 2.2. Sample Collection, DNA Isolation, and Sequencing

The fecal pellets were collected at the time of euthanasia (i.e., at the end of the 24-week intervention period), snap frozen in cryotubes in liquid nitrogen, and placed at −80 °C for storage until processing. Samples were thaw at 4 °C for ~30 min prior to microbial DNA extraction and isolation using a DNeasy PowerSoil Pro Kit (Qiagen, Hilden, Germany) according to the directions provided by the manufacturer. Then, extracted DNA was analyzed for quantity (>10 μL/mL) and quality (a ratio of 260/280 nm > 1.8) with the QIAxpert Instrument (Qiagen, Hilden, Germany) before sending DNA samples to the sequencing center. Bacterial community analysis was performed via next generation sequencing using the MiSeq Illumina platform. Amplicon sequencing of the V4 region of the 16S rRNA gene was performed with the barcoded primer set 515f/806r designed by Caporaso et al. 2011 [23] and following the library preparation protocol provided by the Earth Microbiome Project (EMP) (http://www.earthmicrobiome.org/emp-standard-protocols/ accessed on 1 February 2021). PCR amplifications for each sample were done in triplicate, then pooled and quantified using Accublue^®^ High sensitivity dsDNA Quantitation Kit (Biotium). PCR conditions included an initial denaturation period of 3 min at 94 °C, followed by 35 cycles of denaturation (94 °C), annealing (50 °C) and elongation (72 °C) for 60 s, 60 s, and 105 s, respectively, and a final 10 min hold at 72 °C ending at a continuous hold at 4 °C. A no template control sample was included during the library preparation as a control for extraneous nucleic acid contamination. 200 ng of DNA per sample were pooled and then cleaned using QIA quick PCR purification kit (QIAGEN). The pooled sample concentration was then quantified by Illumina library Quantification Kit ABI Prism^®^ (Kapa Biosystems, Wilmington, MA, USA). Then, the DNA pool was diluted to a final concentration of 4 nM then denatured and diluted to a final concentration of 4 pM with 25% of PhiX. Finally, the DNA library was loaded in the MiSeq Illumina and run using the version 2 module, 2 × 250 paired-end, following the directions of the manufacturer [23]. 

### 2.3. Gut Microbiota Analysis

Taxonomic analysis, bacterial diversity (alpha and beta) analysis, and visualization of the data were performed using Quantitative Insights into Microbial Ecology (QIIME2) version 2021.11. Briefly, sequencing generated a total of 3,715,562 sequences with a mean of 65,185.30 reads per sample. Then, we visually inspected sequence files for quality plots (quality score > 25). Based on 10,000 randomly selected reads, forward reads were trimmed at 6 and truncated at 194 bases and the reverse reads were trimmed at 13 and truncated at 167 bases using the Divisive Amplicon Denoising Algorithm 2 (DADA2) to generate a feature table using amplicon sequence variants (ASVs). Then, the forward and reverse reads were merged into one unique sequence file per sample and chimeric sequences and other errors (e.g., long and short reads) encountered during PCR amplification were removed. Last, the inferred samples were combined into one unified sequence table. We assigned taxonomy to mapping sequences utilizing the q2-feature-classifier plugin and a pre-trained Naïve Bayes classifier using the Greengenes 13_8 99% database. We normalized samples to the maximum sampling depth (15,247) to maintain a high number of counts per sample while maintaining the quality of the sequences for statistical analyses. We created a rooted phylogenetic tree utilizing fasttree and mafft alignment for further investigation of diversity metrics that included phylogeny. 

We assessed alpha diversity (within-group diversity) with richness metrics (Observed ASVs, Shannon diversity index, and Faith’s Phylogenetic diversity) and evenness (Pileous index). For beta-diversity (between-group diversity), qualitative (Jaccard distance and unweighted UniFrac distance), and quantitative (Bray–Curtis dissimilarity and weighted UniFrac distance) metrics were calculated and visualized using Emperor in QIIME2. Relative abundance of taxa for each group was calculated at the genus level. 

### 2.4. Statistical Analyses

The current study was a secondary analysis of a study powered to assess total protein expression in tissues using Western blot, and serum assays which required between 7–10 mice/group. Gut microbiota was a secondary aim and therefore not powered for a priori. Normality of the data was assessed utilizing Shapiro–Wilk tests, and visual inspection of q–q plots. We calculated the weight gain as the difference between the baseline body weight and body weight at 24 weeks. Then, we compared the differences in body weight change and feed/water intake between groups by a one-way ANOVA test followed by Tukey post-hoc pairwise comparison. All group differences of alpha diversity metrics were assessed using Kruskal–Wallis nonparametric tests. Beta-diversity group differences were calculated by Permutational Multivariate Analysis of Variance (PERMANOVA). The linear discriminate analysis of effect size (LefSe) tool was used to compare the differential abundance of microbial taxa across groups. To determine differentially abundant taxa between groups, we first used a non-parametric Kruskal–Wallis test and pairwise Wilcoxon rank-sum test to detect significant differential abundance taxa, followed by a linear discriminate analysis to estimate the effect size of each differentially abundant taxa. Alpha was set at 0.05 for all tests of significance, and the Benjamini–Hochberg correction (q; adjusted *p*-value) was used for multiple comparisons following PERMANOVA analyses. 

## 3. Results

### 3.1. Body Weight Gain and Food Intake

Weight gain assessed over the 24-week study duration was significantly increased in the HFD (36.02 ± 1.28 g) group compared to the chow group (20.5 ± 1.35 g); *p* < 0.05). Switching mice from HFD to chow (HFD + chow) for the second 12 weeks reversed the weight gain to lean-like weights (chow; 23.67 ± 1.28 g). HFD + Gen + Exe significantly reduced weight gain induced by HFD (31.13 ± 0.78 g, *p* < 0.05) but the total weight change did not reach that of chow or HFD + chow (Figure 2). The progression of weight gain is described in Figure 3. We found no differences in the feed, caloric, and water intake among the groups.

### 3.2. Alpha-Diversity

Taxonomic richness was significantly reduced with consumption of HFD for the 24-week period (Figure 4A,B). Returning to a chow diet (HFD + chow) was the only treatment which reverted the effects of the HFD on microbial richness back to that of chow (Figure 4A). We found significant differences in phylogeny between chow and all groups that started on an HFD (Figure 4C) with HFD + Exe and HFD + Gen resulting in significantly lower phylogenetic diversity when compared to HFD (Figure 4C). Bacterial evenness (Pileous evenness, Figure 4D) was not affected by 24-week HFD as we did not find any significant differences between any groups. Overall, neither exercise nor genistein were able to shift the gut microbiota alpha diversity back to that of the chow-fed lean controls. 

### 3.3. Beta Diversity

We found significant (*p* < 0.05) differences in beta diversity between all groups (Appendix A) when comparing microbiota community structure using Jaccard (Figure 5A) and unweighted UniFrac metrics (Figure 5C). However, when evaluating quantitative metrics of beta-diversity some of these pairwise differences were lost. For Bray-Curtis (Figure 5B), HFD + Gen + Exe did not differ significantly from HFD and HFD + Exe suggesting that abundance was not shifted enough to result in differences between these groups. We further determined if phylogeny and abundance was significantly affected by dissimilarities in the presence/absence of microbial taxa between groups (weighted UniFrac; Figure 5D). Significant differences were observed between HFD + chow and all other HFD groups, but not with chow. For this metric, HFD did not show any differences with HFD + Exe, HFD + Gen, and HFD + Gen + Exe. Similar to the Bray–Curtis analysis, no significant differences were observed between HFD + Exe and HFD + Gen + Exe. Overall, each group affected qualitative beta-diversity distinctively, while the quantitative differences appeared to be driven by HFD + chow, chow, and HFD + Gen.

### 3.4. Linear Discriminate Analysis of Effect Size (LEfSe)

To determine if there were any significant taxa driving the differences observed in the gut microbiota, we compared all groups with one another using linear discriminant analysis of effect size (LEfSe). We did not find any differentially abundant taxa among the six groups. 

## 4. Discussion

The overall goal of the current study was to test the capacity of exercise, genistein, and their combined effect to revert the harmful effects of high-fat, high-sugar diets on gut microbial diversity and body weight. We found that a combined exercise and genistein treatment is needed to significantly reduce weight gain relative to HFD, but the change in weight gain was still significantly greater than animals on chow and HFD + chow diets. The shift from an HFD to a chow diet (HFD + chow) was the only treatment to partially recuperate the bacterial richness compared to mice who never received the HFD treatment (lean controls, chow). Genistein and exercise alone significantly changed the bacterial community structure (beta diversity) without any statistically detectable differences in microbial taxa (LEfSe *p* > 0.05) but HFD + Gen + Exe did not result in consistent differences (qualitative but not quantitative metrics) in microbial diversity when compared to HFD alone.

The anti-obesogenic capacity of exercise is well established in the literature. Genistein as an anti-obesity treatment has recently showed promising results [4,6,6,7,11,24]. Less is known about whether the effects of exercise and genistein are mediated associated with the gut microbiota changes. Cross et al., studying ovariectomized rodents supplemented with soy for 28 weeks found that soy supplementation prevented weight and adipose tissue gains regardless of the animals’ low-fitness status [6]. Also, Cross et al., found soy to significantly modify the relative abundance of *Prevotella* contrary to our results which we did not result in differential microbial abundance. It is possible that soy has a preventative effect over gut microbial abundance but not reversion capacity. Exercise and genistein exerted a synergistic effect reverting the HFD-induced body weight gain in our study but this reduction did not reach that of animals switching to chow. More research is required to confirm our results and determine the underlying mechanisms of genistein, exercise, and their synergistic effect as a potential new therapy for the prevention and treatment of obesity and associated comorbidities (e.g., insulin resistance).

The numerous dietary bioactive compounds available in food make it challenging to discern the impact of each of them on gut microbiota. In our study, genistein did not revert bacterial richness back to that of lean controls; rather, genistein resulted in the lowest bacterial richness and phylogenetic diversity of all six treatments. However, the data on the preventive effects of genistein on high-fat, high-sugar-induced gut microbiota changes are explained elsewhere [4,5,6,7,9,10,11,13,21]. Our results might be explained by insufficient power to see differences with genistein or the lack of studies investigating genistein as a mechanism to treat rather than to prevent. The literature has mostly looked at the effects of genistein on weight and metabolic health outcomes in postmenopausal animal models and species (i.e., ovariectomized rats) [6,10,11,12], meaning that genistein might be useful in certain populations to treat obesity and change gut microbiota. The presence of endogenous estrogen may limit the effect of dietary genistein. The lack of studies using soy bioactive compounds to revert metabolic disease-induced changes in gut microbiota does not allow us to make a definitive conclusion regarding genistein. More studies are needed to investigate the mechanism and dose by which genistein exerts its effects on gut microbiota.

Exercise alone could not significantly counteract the HFD-induced decline in microbial richness. With the same exercise protocol as ours, Petriz et al. [25] found that obese adult rats experienced significant increases in the relative abundance of health-associated genera *Ruminoccocus* spp., *Bifidobacterium* spp., and *Lactobacillus* spp. after 4 weeks of aerobic exercise. In diabetic, non-obese mice, 6 weeks of aerobic exercise significantly increased *Bifidobacterium* spp. [26]. Neither of these studies assessed GM diversity. We observed a large variance in alpha-diversity in the HFD + Exe group which may have affected our ability to capture statistical differences. Exercise seems to exert a preventive effect on the high-fat, high-sugar diet-induced gut microbial diversity as we previously found [16,18] that HFD + Exe and HFD + Exe + Gen resulted in greater microbial richness than HFD alone [21]. The underlying mechanism of how exercise changes gut microbiota is still understudied. More research is needed to understand the heterogeneous results of exercise interventions on gut microbiota in states of obesity. Future studies should consider studying multiple factors such as volume and intensity of exercise to define the right exercise dose for gut microbiota positive effects in obesity treatment.

We found beta-diversity differences among all treatment groups but no significant taxa driving the differences between groups. A recent meta-analysis showed that the weight-management effects of exercise are beta-diversity dependent in humans but not necessarily in rodent models [27]. The rodent models showed both altered alpha-diversity and beta-diversity to be linked to a greater change in body weight. In contrast, in humans, only altered beta-diversity was seen to be dependent on weight loss. This conclusion might be due to the lack of consistency in diversity metrics reported in the literature as it is more common to see beta-diversity metrics in human studies and both alpha and beta-diversity in rodent experiments. 

In this study, where genistein and exercise were used to potentially treat mice on an HFD, we found no significant effects of the combined genistein and exercise treatment on the gut microbiome composition; however, body weight changes were observed. No other studies have tried to test genistein and exercise as a combined treatment to revert high-fat, high-sugar-induced changes on gut microbiota, but there is literature showing that a combination of other bioactive components and exercise exhibit a superior preventative effect on gut microbiota than either of the treatments alone. Cellulose nanofiber with voluntary wheel exercise in male mice showed significant improvement in the relative abundance of *Ruminococcaceae* and *Eubacteriaceae* compared to either of the treatments alone [20]. Our previous data [21] suggested that combining exercise and genistein prevents declines in gut bacterial richness. We also found that *Ruminococcus* spp. were significantly increased in the control and combined treatment groups. Both of these studies were preventive compared to the current project, which tested the use of exercise and genistein as treatments after HFD-induced changes had already been initiated. More research is needed to understand the potential benefits of dietary bioactive compounds and exercise on gut microbiota for obesity treatment. 

A major limitation of this study was the inability to evaluate sex differences because only male mice were included. Another limitation was the collection of fecal samples at only the end of each treatment. A potential factor that could have explained our non-statistically significant effects on gut microbiota in the HFD + Gen and HFD + Exe groups could be the sample size, as this was a secondary analysis for a study powered for a different variable. Future studies may benefit from time-series collections and from collecting both cecal and fecal samples to compare microbiota composition across the gastrointestinal tract. As it has been defined in the literature, cecal samples are more representative of the gut microbiota composition than fecal samples [28,29]. A strength of this project was the well-controlled dietary intake and exercise protocols. Another strength of the study was the use of forced exercise which has been shown to be superior to voluntary exercise for studying the effects of exercise on the gut microbiome [30].

## 5. Conclusions

Contrary to our hypothesis, the combination of exercise and genistein did not revert the HFD-induced decline in bacterial richness and abundance of gut microbiota in male C57BL/6 adult mice. However, the combination of exercise and genistein did reduce body weight gain while on an HFD better than either treatment alone. The novelty of this study stands on the use of genistein and the combination of genistein and exercise as a potential lifestyle treatment to revert diet-induced obesity and diet-induced gut microbial changes. Further research is required to confirm our results in a larger sample size including both sexes.

## Figures and Tables

**Figure 1 microorganisms-10-02221-f001:**
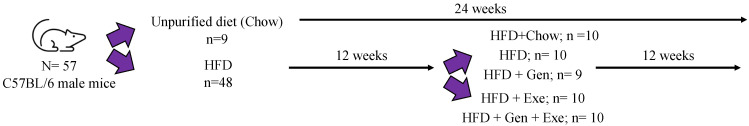
Study Design. High-fat, high-sugar diet, HFD; high-fat, high-sugar diet + genistein (HFD + Gen); high-fat, high-sugar diet + exercise (HFD + Exe); high-fat, high-sugar diet + genistein + exercise (HFD + Gen + Exe). All mice fed with a HFD (20% carbohydrate, 20% protein, and 60% fat) water was supplemented with 42 g/L of sweetener (55% fructose and 45% sucrose). Exe protocol consisted of a 3-week graded increase in exercise duration and intensity as follows: week 1, 10 min at 10 m/min; week 2, 20 min at 10 m/min; week 3, 30 min at 12 m/min; weeks 4–12, 30 min at 15 m/min. This exercise protocol was designed to align with recommendations for cardiovascular health. Gen was included in the mice feed at a concentration of 600 mg of genistein/kg of diet. Purple arrows describe randomization.

**Figure 2 microorganisms-10-02221-f002:**
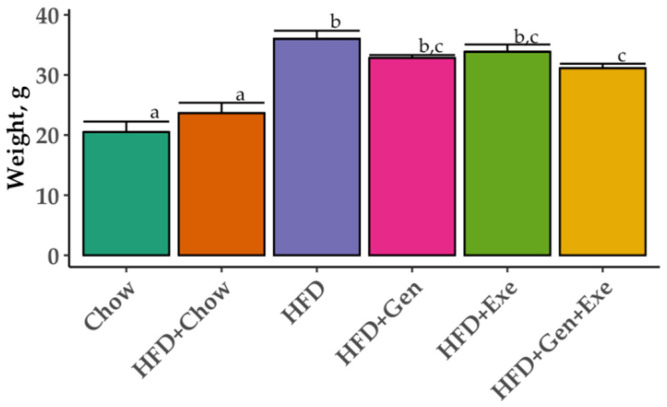
Mean ± SE weight gain (g) over 24 weeks of six different dietary and exercise interventions. Treatment groups are as follows: unpurified diet (chow); high-fat, high-sugar diet + chow (HFD + chow); HFD; HFD + exercise (HFD + Exe); HFD + genistein (HFD + Gen); HFD + Exe + Gen. Significant group differences (*p* < 0.05) by one-way analysis of variance (ANOVA) test after Tukey post-hoc pairwise comparison. Different letters denote significant differences.

**Figure 3 microorganisms-10-02221-f003:**
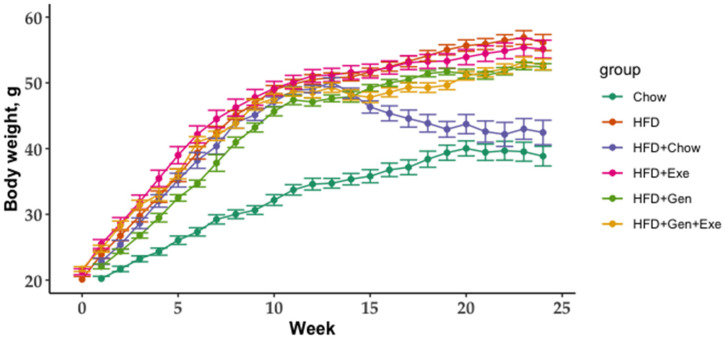
Body weight (g) over 24 weeks of six different dietary and exercise interventions using treatment groups are as follows: unpurified diet (chow); high-fat, high-sugar diet (HFD + chow); HFD; HFD + exercise (HFD + Exe); HFD + genistein (HFD + Gen); HFD + Exe + Gen.

**Figure 4 microorganisms-10-02221-f004:**
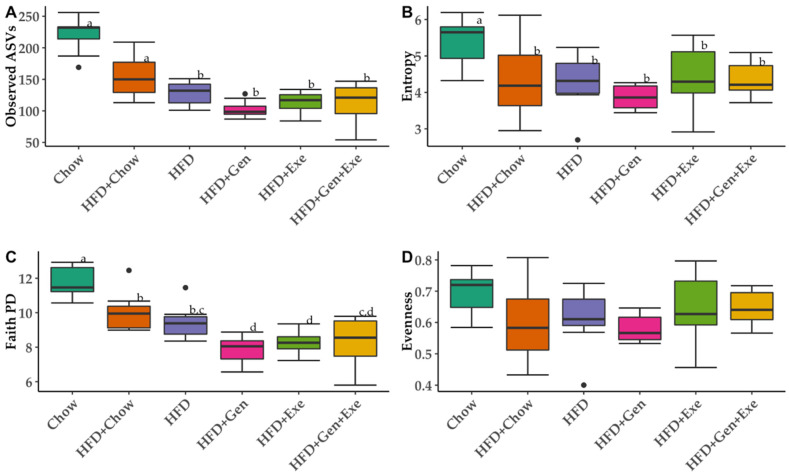
Microbial alpha-diversity (within-group diversity) after 24 weeks of six different dietary and exercise interventions using (**A**) observed amplicon sequence variants (ASVs; bacterial richness), (**B**) Shannon diversity index (entropy; microbial abundance and evenness), (**C**) Faith Phylogenetic diversity index (Faith PD) and (**D**) Pileous Evenness index (evenness) values. Treatment groups are as follows: unpurified diet (chow); high-fat, high-sugar diet (HFD + chow); HFD; HFD + exercise (HFD + Exe); HFD + genistein (HFD + Gen); HFD + Exe + Gen. Significant group differences (*p* < 0.05) by Kruskal–Wallis test after Benjamini–Hochberg correction for multiple comparisons. Different letter denotes significant differences.

**Figure 5 microorganisms-10-02221-f005:**
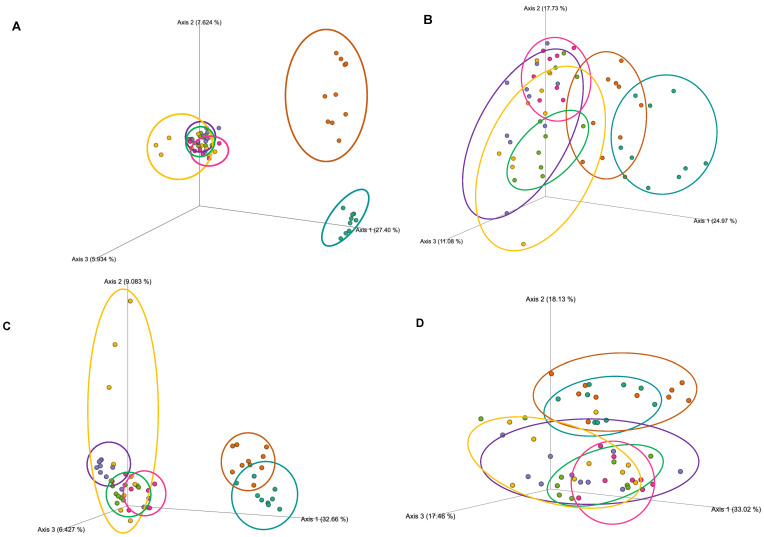
Microbial beta-diversity (between group diversity) PCoA plots representing gut microbiota diversity after 24 weeks following six different dietary and exercise interventions: (**A**) Jaccard distance matrix, (**B**) Bray–Curtis distance matrix values, (**C**) unweighted UniFrac distance matrix, and (**D**) weighted UniFrac distance matrix values. Treatment groups are as follows: unpurified diet (chow), turquoise; high-fat, high-sugar diet + chow (HFD + chow), orange; HFD, purple; HFD + exercise (HFD + Exe), green; HFD + genistein (HFD + Gen), pink; and HFD + Exe + Gen, yellow. Benjamini–Hochberg corrections for multiple comparisons were performed for each comparison.

**Table 1 microorganisms-10-02221-t001:** Customized high-fat diet composition.

Ingredient	Kcal/gm	Grams/kg	Kcal/kg
Casein	3.58	258.5	925.43
Cornstarch	2.6	0	0
Dyetrose	3.8	161.6	614.08
Sucrose	4	88.9	355.60
Cellulose	0	64.6	0
Corn oil	9	32.3	290.70
Lard	9	316.6	2849.40
Salt Mix 210088	1.6	12.9	20.64
Dicalcium Phosphate	0	16.8	0
Calcium Carbonate	0	7.1	0
Potassium Citrate H_2_O	0	21.3	0
Vitamin Mix 30050	3.92	12.9	50.57
L-Cystine	4	3.9	15.60
Choline Bitartrate	0	2.6	15.60

## Data Availability

Not applicable.

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
