# Peer review of "Exercise and/or Genistein Do Not Revert 24-Week High-Fat, High-Sugar Diet-Induced Gut Microbiota Diversity Changes in Male C57BL/6J Adult Mice"

_microorganisms, 2022, doi:10.3390/microorganisms10112221_

Round 1

Reviewer 1 Report

Authors investigate the effect of genisten and exercise in male mice fed obesogenic diets on body weight and gut microbiota. The study is simple and interesting. Currently, the effect of exercise and nutrition on gut microbiota has not been completely explained. However, the study can not be published in the current form. Some points are required to be clarified prior to publication, as listed below.

Abstract

Headings such as background, methods, results and conclusions should be removed.

Line 13: Please specify what is meant for unpurified diet. Acronyms such as HFD should be listed in extenso the first time that they appear in the text.

Line 15: Statistical approach could be removed from the abstract.

Line 18: When presenting statistically significant results the p-value should be provided. In addition, each comparison should be clear e.g. significantly decreased weight gain relative to HDF “compared to…”.

Line 19: Which index or indexes related to richness?

Line 23-24: Revise the English.

Introduction

Line 37: Move the dot after the double brackets of the citation.

Line 38: Correct the typo

Line 41: Specify the area of intestine that was analysed

Line 43: 20-40 mg/kg of feed or BW?

Line 47: correct the typo.

Line 54: Move the dot after the reference

Line 60: Firmicutes and Bacteroidetes should be listed in italic

Line 61: The number of the citation is lacking.

Line 64: Which kind of exercise?

Line 69: Can you define what is meant for “exercise”?

Line 75: Can you provide more information regarding how much cellulose was used in the study?

Line 83: Move the dot after the citation.

Line 87: Can you be more precise concerning the deleterious effects?

Line 88: Why did you select a period of 24 weeks?

Materials and methods

Line 93: Is there a specific reason for selecting only males? Were they balanced per weight? Which was the initial weight of animals?

Line 95: Please provide the authorization number (line 116-117).

Line 96: The extended form of HFD is lacking. What is intended for unpurified diet?

Line 105: Substitute food with feed. Ad libitum should be listed in italic.

Line 106: Can you provide the relative humidity?

Line 110: This sentence is incomplete. You should report all ingredients in a decreasing order or reference the composition to table 1 without listing ingredients.

Line 114-116: This phrase could be used at the beginning of this chapter.

Figure 1: Add in the figure caption the extended form for all acronyms.

Line 122-123: How did you select the following protocol of exercise?

Table 1: Why corn starch was included at 0 g/kg? Is it useful to report this ingredient in the list? Does cellulose have 0 kcalories? Can you provide the nutrient composition of unpurified diet?

Line 128: Clarify when you euthanized animals (at which timepoint).

Line 131: Did you evaluate the quality of extracted DNA?

Line 136: Please provide the PCR conditions.

Line 146: Move the dot after the reference.

Line 177: Did you evaluate data normality? Substitute food with feed.

Line 178: Have you considered repeated measurements?

Line 183: Correct the English.

Results

Use weight gain only referring to average daily gain (g/day) and not for body weight (g)

Line 194: Include averages ± error range of both groups in order to clarify the observed difference and provide the p-value for significant differences.

Line 195: Weight gain was reduced compared to? Please include also value from the other group.

Line 197: Do not report non-significant p-values.

Line 210-218: Provide p-values for statistically significant results.

Line 255: Remove non-significant p-value

Line 268: Same comment as before

Discussion

Line 273: The citation should be placed after Cross et al. [6].

Line 295: Do you believe that your treatment can be effective only in males and after menopausal?

Line 303: Spp. should be listed without using italic. Correct it in this section.

Line 355: Revise the English.

Author Response

Dear reviewer, 

Thank you for taking the time to review our manuscript and for the detailed feedback. Please, find below our answers to your concern and recommended changes addressed. 

Reviewer 1

Comment

Answer

Headings such as background, methods, results and conclusions should be removed.

Headings in abstract have been removed

Line 13: Please specify what is meant for unpurified diet. Acronyms such as HFD should be listed in extenso the first time that they appear in the text.

The acronym of HFD is defined before the use of HFD. We described unpurified diet as the regular chow diet.

Line 15: Statistical approach could be removed from the abstract.

We have made this edit.

Line 18: When presenting statistically significant results the p-value should be provided. In addition, each comparison should be clear e.g. significantly decreased weight gain relative to HDF “compared to…”.

We add the p values and the changes are relative to the HFD group. Please see, HFD+Exe+Gen significantly (p<0.05) decreased weight gain relative to HFD with only HFD+chow reverting the body weight change to that of chow.

Line 19: Which index or indexes related to richness?

The richness index is updated

Line 23-24: Revise the English.

Please see edit on last lines of abstract.

Line 37: Move the dot after the double brackets of the citation.

All citations have been revised and updated to have the dot after the citation.

Line 41: Specify the area of intestine that was analysed

Updated the introduction.

Line 43: 20-40 mg/kg of feed or BW?

20-40 mg of genistein/kg of body weight

Line 60: Firmicutes and Bacteroidetes should be listed in italic

Updated the formatting of the microbes to italic.

Line 61: The number of the citation is lacking

Citation was updated

Line 64: Which kind of exercise

The exercise type was updated

            Line 69: Can you define what is meant for “exercise”?

The exercise was updated.

Line 75: Can you provide more information regarding how much cellulose was used in the study?

Amount of cellulose was updated.

Line 87: Can you be more precise concerning the deleterious effects?

We have amended this sentence to articulate what adverse effects were expected to be improved.

Line 88: Why did you select a period of 24 weeks?

The analysis of the gut microbiota was an exploratory aim from a parent study from which the selection of the length of the intervention was selected. We mentioned in line 204: The current study was a secondary analysis of a study powered to assess total protein expression in tissues using western blot, and serum assays which required between 7-10 mice/group. Gut microbiota was a secondary aim and therefore not powered for a priori.

We chose 24 weeks since - a 12-week duration is adequate to get all mice obese and diabetic and at a new steady state of diabetic obesity: the following 12 weeks was aimed to provide enough time for observing/evaluating potential reversal effects of the diabetic obesity.  This is validated in the switch from HFHS to standard chow/water, whereby body weight reverses to lean control levels in that second 12 weeks, as we approach 24 week study duration.

Line 93: Is there a specific reason for selecting only males? Were they balanced per weight? Which was the initial weight of animals?

We chose males since for our 12 week prevention study  males responded very well to HFHS and females were a bit ore variable in weight changes, developing hyperglycemia. Aslo, the use of only males is due to a lack of funding to purchase a bigger set of mice with both sexes (female and male) and we are planning for a larger cohort in the future with male and female mice. Mice weight was balanced as presented in the figure 3. The initial body weight of the animals was not significantly different across groups.

Line 95: Please provide the authorization number (line 116-117).

The authorization number was provided in line 147-148

Line 96: The extended form of HFD is lacking. What is intended for unpurified diet?

The utilization of the unpurified diet is to serve as a control for the entire duration for the 24 weeks.

Line 105: Substitute food with feed. Ad libitum should be listed in italic

The word ad libitum was updated to be italic

Line 106: Can you provide the relative humidity?

Updated in the text.

Line 110: This sentence is incomplete. You should report all ingredients in a decreasing order or reference the composition to table 1 without listing ingredients.

We have adjusted this sentence.

Figure 1: Add in the figure caption the extended form for all acronyms.

All the acronyms were updated.

Line 122-123: How did you select the following protocol of exercise?

The exercise protocol selected is equivalent to the recommended cardiovascular physical activity and exercise in humans for cardiovascular health per the guidelines from AHA for moderate exercise 150 min /week.

Line 128: Clarify when you euthanized animals (at which timepoint).

Clarification when the animals were euthanized was updated

Line 131: Did you evaluate the quality of extracted DNA?

Yes. A sentence explaining the evaluation of the DNA was added.

Line 136: Please provide the PCR conditions

PCR conditions were based on the Earth Microbime Protocol and included an initial denaturation period of 3 min at 94C, followed by 35 cycles of denaturation (94C), annealing (50C) and elongation (72C) for 60 s, 60 s, and 105 s, respectively, and a final 10 min hold at 72C ending at a continuous hold at 4C.

Line 177: Did you evaluate data normality? Substitute food with feed.

We assessed normality of the data. A sentence with the methods was added. We updated from food to feed.

Line 178: Have you considered repeated measurements?

We collected samples post intervention; therefore we did not consider repeated measures as we did not have baseline data to perform such analyses. it is an interesting point. And could be a future study. I think overall having the lean controls and HFHS as the western diet controls both maintained on those diets for 24 weeks serves well as a comparison for the diet/exercise groups that were changed 12 weeks into the study.

Use weight gain only referring to average daily gain (g/day) and not for body weight (g)

We updated the name as suggested.

Line 194: Include averages ± error range of both groups in order to clarify the observed difference and provide the p-value for significant differences.

Error bars now appear on Figure 2 and p-values have been denoted in the figure legend and in the corresponding text.

Line 195: Weight gain was reduced compared to? Please include also value from the other group.

Please see the updated sentences lines 231-234

Line 197: Do not report non-significant p-values. Line 210-218: Provide p-values for statistically significant results.

The p values were updated.

Line 295: Do you believe that your treatment can be effective only in males and after menopausal?

With the data collected in this study, we are not able to answer this question. The treatment has been predominantly used in estrogen-positive cancer treatment to treat postmenopausal symptoms. The utilization of genistein as an anti-obesogenic effect is novel and requires further research with different amounts of isoflavone and different exposure lengths. Exercise has shown to be a great intervention to prevent and revert obesity; however, there are also exercise nonresponders that remain an intense area of research. Combining bioactive ingredients and exercise as an anti-obesogenic treatment to revert obesity through gut microbiota is a field that will require further investigation. Future studies should be powered to detect changes in gut microbiota as well as analyze fecal, cecum, plasma, and adipose tissue gut-derived metabolites to enhance our understanding of how bioactive ingredients might play a role in weight management.

Line 303: Spp. should be listed without using italic. Correct it in this section.

The spp was updated as recommended.

Line 355: Revise the English.

Updated.

Reviewer 2 Report

The present manuscript focuses on the Exercise and/or Genistein Do Not Revert 24-week High-Fat High-Sugar Diet-Induced Gut Microbiota Diversity Changes in Male C57BL/6J Adult Mice.  The subject frame of the work is well constructed. So, in this respect and this article should be contributed to present research. I recommended this work for publication after the following minor revisions.

1.      There are several typographical mistakes as well in whole manuscript. Therefore, the author’s thoroughly careful check the language and typo mistake to minimize the error.

2.      The abstract should be beginning with a sentence about the background of concept and the aims as well as novelty of study should be mentions. What exactly is the novelty of this study? The abstract is poorly written and should be improved. Abbreviations must be avoided in abstract. Parenthesis should be avoided in abstract - this is poor writing. Please improve.

3.      The introduction and discussion section about microbiota need extensive revision and improved. Be specific and adhere to importance of topic.

4.      All figures are of poor technical quality and not suitable for publication, especially in a high reputed journal. Font size and kind is too small and must be unified in all figures. Small writings are unreadable. All figures must be self-explanatory. Axis titles are poorly presented or absent. Units are missing. Are the data presented in figures significantly different? At least error bars should be shown.

5.      What is exactly the novelty of this review article, as so many articles were already out, is this the updates version or some other novel aspect. Author needs to revised it carefully and should provide novelty statement.

6.      Bacteria species/genus name should be italic throughout the manuscript, for example Enterococcus in all sub heading etc. 

Author Response

Dear reviewer, 

Thank you for taking the time to review our manuscript and for the detailed feedback. Please, find below our answers to your concern and recommended changes addressed. 

Reviewer 2

2.      The abstract should be beginning with a sentence about the background of concept and the aims as well as novelty of study should be mentions. What exactly is the novelty of this study? The abstract is poorly written and should be improved. Abbreviations must be avoided in abstract. Parenthesis should be avoided in abstract - this is poor writing. Please improve.

The abstract was updated including the suggested aim of the study, novelty, and background.

3.      The introduction and discussion section about microbiota need extensive revision and improved. Be specific and adhere to importance of topic.

The literature was updated as suggested

4.      All figures are of poor technical quality and not suitable for publication, especially in a high reputed journal. Font size and kind is too small and must be unified in all figures. Small writings are unreadable. All figures must be self-explanatory. Axis titles are poorly presented or absent. Units are missing. Are the data presented in figures significantly different? At least error bars should be shown.

Figures are updated considering the suggested comments, please see new figures.

5.      What is exactly the novelty of this review article, as so many articles were already out, is this the updates version or some other novel aspect. Author needs to revised it carefully and should provide novelty statement.

The novelty statement has been included in the conclusion section. There is growing literature on the effects of exercise on the gut microbiota. However, there is not much literature investigating the potential benefits of targeting the gut microbiota through nutrition + exercise interventions. Our aim was to explore the potential effects of the synergistic effect of genistein and exercise to recuperate the high-fat, high-sugar diet-induced gut microbiota diversity changes as the reduction and changes in gut microbial diversity have been associated with obesity and comorbidities.